# Tafenoquine following G6PD screening versus primaquine for the treatment of vivax malaria in Brazil: A cost-effectiveness analysis using a transmission model

David J. Price[ORCID][1,2], Narimane Nekkab[ORCID][3,4], Wuelton M. Monteiro[ORCID][5,6], Daniel A. M. Villela[ORCID][7], Julie A. Simpson[ORCID][2], Marcus V. G. Lacerda[ORCID][5,8], Michael T. White[ORCID][9], Angela Devine[ORCID][2,10,11]*

1 Department of Infectious Diseases, The University of Melbourne, at the Peter Doherty Institute for Infection and Immunity, Victoria, Australia, 2 Centre for Epidemiology and Biostatistics, Melbourne School of Population and Global Health, The University of Melbourne, Melbourne, Victoria, Australia, 3 Swiss Tropical and Public Health Institute, Basel, Switzerland, 4 University of Basel, Basel, Switzerland, 5 Instituto de Pesquisa Clínica Carlos Borborema, Fundação de Medicina Tropical Dr Heitor Vieira Dourado, Manaus, Brazil, 6 Escola Superior de Ciências da Saúde, Universidade do Estado do Amazonas, Manaus, Brazil, 7 Programa de Computação Científica, Fundação Oswaldo Cruz, Rio de Janeiro, Brazil, 8 Instituto Leônidas & Maria Deane–ILMD, Fundação Oswaldo Cruz, Manaus, Brazil, 9 Institut Pasteur, Université de Paris, G5 Épidémiologie et Analyse des Maladies Infectieuses, Département de Santé Globale, F-75015 Paris, France, 10 Global and Tropical Health Division, Menzies School of Health Research, Charles Darwin University, Darwin, Northern Territory, Australia, 11 Melbourne Health Economics, Centre for Health Policy, Melbourne School of Population and Global Health, The University of Melbourne, Melbourne, Victoria, Australia

* angela.devine@menzies.edu.au

## Abstract

### Background

Malaria transmission modelling has demonstrated the potential impact of semiquantitative glucose-6-phosphate dehydrogenase (G6PD) testing and treatment with single-dose tafenoquine for *Plasmodium vivax* radical cure but has not investigated the associated costs. This study evaluated the cost-effectiveness of *P. vivax* treatment with tafenoquine after G6PD testing using a transmission model.

### Methods and findings

We explored the cost-effectiveness of using tafenoquine after G6PD screening as compared to usual practice (7-day low-dose primaquine (0.5 mg/kg/day) without G6PD screening) in Brazil using a 10-year time horizon with 5% discounting considering 4 scenarios: (1) tafenoquine for adults only assuming 66.7% primaquine treatment adherence; (2) tafenoquine for adults and children aged >2 years assuming 66.7% primaquine adherence; (3) tafenoquine for adults only assuming 90% primaquine adherence; and (4) tafenoquine for adults only assuming 30% primaquine adherence. The incremental cost-effectiveness ratios (ICERs) were estimated by dividing the incremental costs by the disability-adjusted life years (DALYs) averted. These were compared to a willingness to pay (WTP) threshold of US$7,800 for Brazil, and one-way and probabilistic sensitivity analyses were performed.

**Data Availability Statement:** The transmission model code is available at https://github.com/MWhite-InstitutPasteur/Pvivax_TQ_IBM.

Parameters for the economic analyses are described in the manuscript and supplementary materials.

**Funding:** This work was supported by Medicines for Malaria Ventures. WMM is funded by FAPEAM through POSGRAD and Pró-Estado public calls. WMM and ML are fellows of the National Council for Scientific and Technological Development (CNPq). MTW is funded by The Bill & Melinda Gates Foundation [grant number INV-024368]. The sponsor had no role in the study design, analysis, decision to publish, or preparation of the manuscript.

**Competing interests:** The authors have declared that no competing interests exist.

**Abbreviations:** DALY, disability-adjusted life year; G6PD, glucose-6-phosphate dehydrogenase; ICER, incremental cost-effectiveness ratio; PSA, probabilistic sensitivity analysis; SIVEP, Malaria Epidemiological Surveillance Information System; US$, United States Dollars; WHO, World Health Organisation; WTP, willingness to pay; 95% CrI, 95% credible interval.

All 4 scenarios were cost-effective in the base case analysis using this WTP threshold with ICERs ranging from US$154 to US$1,836. One-way sensitivity analyses showed that the results were most sensitive to severity and mortality due to vivax malaria, the lifetime and number of semiquantitative G6PD analysers needed, cost per malaria episode and per G6PD test strips, and life expectancy. All scenarios had a 100% likelihood of being cost-effective at the WTP threshold. The main limitations of this study are due to parameter uncertainty around our cost estimates for low transmission settings, the costs of G6PD screening, and the severity of vivax malaria

## Conclusions

In our modelling study that incorporated impact on transmission, tafenoquine prescribed after a semiquantitative G6PD testing was highly likely to be cost-effective in Brazil. These results demonstrate the potential health and economic importance of ensuring safe and effective radical cure.

## Author summary

### Why was this study done?

- Radical cure with primaquine or recently approved tafenoquine is required to clear the dormant liver parasites of vivax malaria.

- While single-dose tafenoquine overcomes the barrier of patient adherence to the current 7-day primaquine treatment, it costs more and requires screening for glucose-6-phosphate dehydrogenase (G6PD) deficiency.

- While the impact of changing policies to tafenoquine after G6PD screening on transmission has been evaluated, the associated costs and cost-effectiveness will be important considerations for policymakers.

### What did the researchers find?

- Using an economic evaluation model coupled with a transmission model, we found that prescribing tafenoquine to vivax malaria patients without G6PD deficiency would be highly likely to be cost-effective in Brazil.

- Tafenoquine will be particularly cost-effective in settings where patient adherence to the current 7-day treatment is low and when paediatric tafenoquine is available to treat children as well as adults.

### What do these findings mean?

- To our knowledge, this is the first study that has looked at the cost-effectiveness of tafenoquine when including the impact on disease transmission.

- The high probability of cost-effectiveness across a wide range of scenarios and munici-palities should reassure decision-makers in Brazil, where tafenoquine has recently been adopted into national policy, and aid other countries considering the implementation of tafenoquine after G6PD screening.

## Background

The burden of malaria in Brazil is primarily due to *Plasmodium vivax*, with an estimated 168,499 indigenous *P. vivax* cases reported in 2018 [1,2], contributing to an estimated national societal cost of 17.6 million United States Dollars (US$) in 2017 [3]. The control of vivax malaria is more challenging than *P. falciparum* since the parasite forms hypnozoites, dormant liver parasites that can cause multiple relapsing episodes of malaria and ongoing transmission. Current practice in Brazil is to prescribe radical cure with a 3-day treatment of chloroquine for the blood-stage parasites and 7-day low-dose (3.5 mg/kg total) primaquine to kill the liver-stage parasites. Full adherence to this regimen is suboptimal in South America, with studies estimating adherence ranging from 62% to 86% [4–8]. These estimates are likely to be elevated since observer bias results in an increase in a patient's likelihood to adhere to a full course of treatment [9].

Recent clinical trials have demonstrated that tafenoquine has comparable efficacy to low-dose primaquine (total dose 3.5 mg/kg) but has the advantage of being administered as a single dose [10,11]. Patients with vivax malaria who have glucose-6-phosphate dehydrogenase (G6PD) deficiency, an inherited enzymopathy, are at risk of drug-induced haemolysis when taking either primaquine or tafenoquine. The World Health Organisation (WHO) recommends screening for G6PD deficiency before prescribing primaquine with a threshold for prescribing the drug set at 30% G6PD activity, a level amenable to qualitative diagnostics [12]. The threshold for G6PD deficiency for prescribing tafenoquine, however, is set at 70%, which requires semiquantitative screening in order to exclude individuals with intermediate G6PD deficiency (30% to 70% activity). Current practice in Brazil does not require screening for G6PD deficiency, but this has been associated with hospitalisation and mortality due to prima-quine-induced haemolysis in patients with G6PD deficiency in Brazil [13–15]. Recently updated malaria guidelines in Brazil require G6PD screening before administering primaquine at health facilities that have the capacity to provide G6PD testing [16]. The STANDARD G6PD Test (SD Biosensor, Republic of Korea) has demonstrated good diagnostic accuracy [17] and operational feasibility [18].

The impact of tafenoquine following G6PD screening on transmission in Brazil has been explored in a recent mathematical model [19]. The model simulations demonstrated a decrease in transmission over a 10-year time horizon, although this was not sufficient to achieve elimination. Since a paediatric formulation of tafenoquine is being developed, a scenario is included to evaluate the impact of prescribing to adults and children over the age of 6 months in addition to an adults-only scenario. Here, we conduct a complementary economic evaluation with the transmission model predictions to explore the impact on the overall cost-effectiveness of a semi-quantitative G6PD test-and-treat strategy using tafenoquine for radical cure.

## Methods

Extending the analysis of 10-year projections of vivax malaria cases from a previously published individual-based *P. vivax* transmission model for Brazil [19], we conducted a cost-

effectiveness analysis to estimate the impact of tafenoquine following G6PD screening with a semiquantitative test on costs and disability-adjusted life years (DALYs). A health economic analysis plan was not developed for this model-based analysis.

## Transmission model

Full details of the transmission model can be found in Nekkab and colleagues' study [19] and White and colleagues' study [20]. Briefly, the individual-based transmission model originally calibrated to epidemiological data from Papua New Guinea represents *P. vivax* transmission and includes sources of heterogeneity such as individual-level treatment, seasonality in exposure, and age-specific immunity. The model presented in White and colleagues' study was extended in Nekkab and colleagues' study to the Brazilian context, to capture the 2 predominant modes of transmission: peri-domestic transmission within and near households; and occupational exposure predominantly affecting working age males. The model was extended to capture these modes of transmission with the following: the stratification of individuals in the model as male or female; males further stratified according to their exposure source (peri-domestic or occupational); and 2 populations of mosquitoes representing peri-domestic or occupational exposure. The model was calibrated to the Brazilian context using several data sources from Brazil and validated by the National Malaria Control Programme and malaria experts in Brazil. The model was calibrated to case notification data for 126 municipalities for which there were at least 100 cases in 2018, such that heterogeneity in demographics and transmission intensity was represented. The intervention simulations were generated for a population size of 100,000 and subsequently converted to the appropriate population size for each municipality estimated for 2018. The results cover a 10-year time horizon from 2020 to 2029 and assume that G6PD screening is fully rolled out in January 2021.

The cost-effectiveness analysis presented herein uses the median value of multiple disease and treatment model states across 100 simulations of the individual-based model over time for each municipality. Specifically, the number of primaquine and tafenoquine doses, cases in males and females over 16 years, cases in males and females under 16 years, number of cases in pregnant women, and G6PD tests were provided by the transmission model. The model assumed a truncated exponential age distribution with a mean age of 32 years. Under this model, 15.6% of those under 16 years would be under the age of 2 years and, therefore, be ineligible for tafenoquine. Approximately 4% of those under 16 years would be under the age of 6 months and not receive radical cure, while the remainder of children under 2 years would receive primaquine.

## Treatment scenarios

The original transmission model analysis explored 6 treatment policy scenarios for tafenoquine following G6PD screening [19]; each of which were compared to the baseline scenario of 7-day low-dose primaquine (3.5 mg/kg total) for all eligible patients without G6PD screening. Here, we explore the cost-effectiveness of 4 of those scenarios. Scenario 1 explored tafenoquine for adults over the age of 16 (tafenoquine for adults), while Scenario 2 expanded access to children who were over the age of 2 years (tafenoquine for all). While Scenarios 1 and 2 assumed that adherence to the 7-day primaquine regimen was 66.7% [8], Scenarios 3 (high primaquine adherence) and 4 (low primaquine adherence) explored the impact of assuming 90% and 30% preexisting primaquine adherence when prescribing tafenoquine to adults, respectively. The baseline adherence rate estimate of 66.7% was chosen from a study that appeared to best reflect the current situation in Brazil [8]. The values of 30% and 90% in Scenarios 3 and 4

were selected to be extreme for the purposes of the sensitivity analyses for the transmission model [19].

In all scenarios, screening for G6PD deficiency was done via semiquantitative testing, repeated at each presentation with malaria. Children over the age of 2 years received a G6PD test before being prescribed radical cure. Cases among children were assumed to be proportionally distributed; i.e., we assumed that 4% of cases in children under 16 years occurred in children under the age of 6 months and allocated these cases accordingly as excluded from radical cure. Pregnant and lactating women were not eligible for radical cure, so these women were not tested for G6PD deficiency or assigned any costs for radical cure. Pregnancy resulted in a 9-month exclusion from radical cure, and a further 6 months were excluded for breast-feeding. The total number of G6PD tests in a given municipality and year were divided proportionally in those who were eligible for radical cure by cases in males and females over 16 years and cases in males and females under 16 years.

G6PD deficiency varied by province and G6PD activity score distribution was based on Markov chain Monte Carlo fitted Gaussian mixture models to 4 datasets [19]. For 2 states in which survey data were not available a national-level prevalence of 5.52% G6PD deficiency was used. The diagnostic accuracy of the test was taken from a study in Brazil that was used to inform the US FDA submission for the SD Biosensor STANDARD G6PD Test (Table 1) [21]. Parameters relating to risk of severe malaria, haemolysis, and mortality are provided in Table 1.

## Costs

Costs from a healthcare provider perspective are reported in 2020 US$. When applicable, costs were inflated to 2020 using gross domestic product deflators [26] and converted from Brazilian reais to US$ [27]. Table 1 provides the unit costs and further information about how these costs were applied. The cost of radical cure was also included for severe cases, but it was assumed this would be prescribed after release from the hospital. The semiquantitative machine was assumed to have a lifetime of 5 years (range: 3 to 10). The number of healthcare facilities in each municipality was calculated from the 2018 Malaria Epidemiological Surveillance Information System (SIVEP) data on number of health units that saw at least 100 cases of vivax malaria [2,23]. In the base case, it was assumed that, on average, 1.05 machines (range: 1 to 2) would be needed per healthcare facility.

## DALYs

DALYs for each scenario were calculated by adding the years of life lost to the years of life with disability. Model parameters for years of life with disability are shown in Table 1. It was assumed that all malaria cases resulted in anaemia. The average age of patients with vivax malaria by sex were derived from the SIVEP database [23]. These ages matched to the life expectancy for that age and sex to calculate the years of life lost [28]. Life expectancy for both sexes was varied by ±10% in the sensitivity analyses.

## Analyses

Both costs and outcomes were discounted at 5% per year in the base case analysis to reflect the value that society attaches to present consumption as opposed to consumption in the future in Brazil. In addition, results are also presented for 0% and 10% discounting. Total costs and DALYs were calculated for each scenario as compared to the baseline scenario. The incremental cost-effectiveness ratio (ICER) was calculated by dividing the difference in costs by the number of DALYs averted. The base case analysis was run for each of 124 municipalities in

**Table 1.  Parameters used in the economic evaluation, including diagnostic accuracy of G6PD screening; risk of haemolysis, severe malaria and death; unit costs in 2020 US$; and length of illness and weights for DALYs.**

| Parameter | Base case | Lower | Upper | Distribution | Par. 1 * | Par. 2 † | Source |
|---|---|---|---|---|---|---|---|
| Sensitivity for severe G6PD deficiency (<30%) | 0.999 | 0.94 | 1 | Beta | 40.049 | 0.04009 | [21] |
| True intermediates that are misclassified as <30% activity (no radical cure) with G6PD test | 0.49 | 0.39 | 0.58 | Beta | 51.439 | 53.54 | [21] ±20% |
| True intermediates that are misclassified as ≥70% activity (prescribed tafenoquine) with G6PD test | 0.06 | 0.03 | 0.09 | Beta | 14.98 | 234.7 | [21] ±50% |
| G6PD test specificity for ≥70% activity | 0.95 | 0.9 | 1 | Beta | 79.734 | 4.197 | [21] |
| Lifetime of semiquantitative machine | 5 | 3 | 10 | Normal‡ | 5 | 2.96 | Assumption |
| Number of semiquantitative machines per facility | 1.05 | 1 | 2 | Normal | 1.05 | 0.2551 | Assumption |
| Risk of haemolysis if severe G6PD deficient and given radical cure | 0.038 | 0.015 | 0.061 | Beta | 10.693 | 270.7 | [14] |
| Risk of haemolysis if intermediate G6PD deficient and given radical cure | 0.031 | 0.001 | 0.038 | Beta | 15.069 | 471 | Assumption |
| Mortality due to radical cure-induced haemolysis | 0.011 | 0.005 | 0.016 | Beta | 16.569 | 1490 | [14] |
| Risk of severe malaria requiring hospitalisation | 0.03 | 0.015 | 0.045 | Beta | 15.541 | 502.5 | [13] |
| Risk of mortality due to vivax malaria | 0.0003 | 0 | 0.0005 | Beta | 7.147 | 23820 | [13] |
| Cost of clinical malaria visit | 17 | 9 | 26 | Gamma | 15.667 | 1.085 | [22] ±50% |
| Cost of hospitalisation for severe malaria | 59 | 51 | 66 | Gamma | 240.473 | 0.2454 | [23] |
| Cost of hospitalisation for haemolytic event | 87 | 49 | 124 | Gamma | 21.534 | 4.04 | [23] |
| Cost of primaquine treatment | 0.43 | 0.35 | 0.52 | Gamma | 97.943 | 0.00439 | From Ministry of Health |
| Cost of tafenoquine treatment | 1.78 | 1.42 | 2.09 | Gamma | 110.714 | 0.01608 | [24] |
| Cost of semiquantitative test machine | 619 | 375 | 688 | Gamma | 69.487 | 8.908 | From test distributor |
| Cost of semiquantitative test strip | 6.8 | 4.1 | 8.2 | Gamma | 47.074 | 0.1445 | From test distributor (Assumes no wastage) |
| Cost of monthly quality assurance | 20 | 17 | 23 | Gamma | 171.488 | 0.1166 | Cost of controls from test distributor. Cost is divided by the number of patients per month. |
| Cost for annual healthcare worker training | 58 | 29 | 87 | Gamma | 16.103 | 3.602 | Includes room rental, staff time, stationery, and catering. Assumed 2 healthcare workers per healthcare facility. |
| Cost per blood draw for G6PD screening | 0.64 | 0.32 | 0.96 | Gamma | 16.103 | 0.03975 | Local data ±50%. An additional blood draw is needed per person screened for G6PD deficiency |
| Length of disability for severe malaria (fraction of year) | 0.00821918 | 0.002740 | 0.01918 | Beta | 3.384 | 408.4 | Assumption. 3 days with a range of 1–7 days. |
| Length of disability for severe malaria (fraction of year) | 0.01917808 | 0.008212 | 0.02740 | Beta | 16.791 | 858.7 | Assumption. 7 days with a range of 3–10 days. |
| Length of disability for anaemia due to clinical malaria (fraction of year) | 0.08333333 | 0.04167 | 0.1667 | Beta | 5.394 | 59.33 | Assumption. 1 month with a range of 0.5–2 months. |
| Length of disability for anaemia due to severe malaria (fraction of year) | 0.25 | 0.08333 | 0.5 | Beta | 3.592 | 10.78 | Assumption. 3 month with a range of 1–6 months. |
| Disability weight for clinical malaria | 0.051 | 0.032 | 0.074 | Gamma | 22.542 | 0.002262 | [25] |
| Disability weight for severe malaria | 0.133 | 0.088 | 0.190 | Gamma | 25.732 | 0.005169 | [25] |
| Disability weight for moderate anaemia due to vivax malaria | 0.052 | 0.034 | 0.076 | Gamma | 22.991 | 0.002262 | [25] |
| Disability weight for severe anaemia due to severe malaria or haemolysis | 0.149 | 0.101 | 0.209 | Gamma | 28.854 | 0.005164 | [25] |
| Female life expectancy^ | - | - | - | Normal | - | - | [28] |
| Male life expectancy^ | - | - | - | Normal | - | - | [28] |

*Represents shape 1 for beta, mean for normal and truncated-normal, and shape for gamma.

†Represents shape 2 for beta, standard deviation for normal and truncated-normal, and scale for gamma.

‡Normal truncated to [1,∞), on the assumption that the machines do not fail in the first year of their life.

^Base case varied by ±10% in the sensitivity analyses.

DALY, disability-adjusted life year; G6PD, glucose-6-phosphate dehydrogenase; US$, United States Dollars.

Brazil that reported at least 100 vivax malaria cases to SIVEP during 2018. (Note that this excludes 2 municipalities with low cases included in Nekkab and colleagues' study [19].) Two municipalities representing low transmission settings were excluded from the analysis (Ji-Paraná and Paragominas) on the basis that stochastic variability and fadeout (i.e., case counts reaching zero as a result of fluctuations at low case numbers, and in the absence of importation) dominated the low numbers of cases in the transmission model simulations. These results were mapped to show variation between municipalities, and the overall ICER is presented as the national cost divided by the national DALYs averted. A willingness to pay (WTP) threshold of US$7,800 (40,000 Brazilian reais) was used. This is the threshold set by the National Commission for the Incorporation of Technologies (CONITEC) and represents the maximum value that an intervention should cost per DALY averted to be considered cost-effective [29]. CONITEC is the health technology assessment agency for the Brazilian Unified Health System.

A one-way sensitivity analysis and probabilistic sensitivity analysis (PSA) was run on all parameters in the cost-effectiveness analyses. Table 1 contains a list of all parameters, with the point estimate used as base case, and lower and upper values used directly in the one-way sensitivity analysis. For the PSA, the point estimate, lower and upper values were used to fit appropriate distributions from which to sample (i.e., by matching the lower and upper values to 2.5 and 97.5 percentiles, respectively). Gamma distributions were used for costs and DALY weights, and beta distributions were used for all other parameters except for the lifetime of the semiquantitative machine. For this parameter, we assumed a normal distribution truncated below at 1 year, on the assumption that the average semiquantitative device lifetime was at least 1 year. A range of 2 to 10 years was used for the lifetime of the semiquantitative machine in the one-way sensitivity analysis. Table 1 also contains the 2 fitted distributional model parameters for each cost-effectiveness analysis model parameter. A total of 10,000 model parameters were sampled from the specified distributions. The PSA model parameters were each applied to the transmission model output for each municipality for the 4 scenarios, enabling the overall mean costs, DALYs, and ICERs to be calculated and the 2.5 and 97.5 percentiles (referred to herein as the 95% credible interval [95% CrI]).

To provide insights into a range of epidemiological transmission settings, the results of the sensitivity analyses are presented for three municipalities: peri-urban Manaus, São Gabriel da Cachoeira, and Itaituba. The first 2 municipalities are in the state of Amazonas, while Itaituba is in Pará state. Peri-urban Manaus excludes the malaria-free urban areas of that municipality. These municipalities were selected as exemplars of where occupational exposure (Itaituba) or peri-domestic transmission (Sao Gãbriel da Cachoeira) were the dominant modes of transmission, or a combination of both modes (peri-urban Manaus). Transmission intensity was 23 cases per 1,000 person years in Itaituba, 114/1,000 in peri-urban Manaus, and 267/1,000 in São Gabriel da Cachoeira [19].

The model code is available in S1 Appendix.

### Ethics approval and consent to participate

Not applicable.

## Results

### Base case analysis

The base case results for using tafenoquine for *P. vivax* treatment in Brazil after G6PD screening with 5% discounting were consistently cost-effective using a WTP threshold of US$7,800 (Table 2). The ICER for Scenario 1 (tafenoquine for adults) was US$982, dropping to US$472

**Table 2. Total overall cases, costs, DALYs, and ICERs for Brazil from 2020–2029.**

| Scenario | Cases | Cases averted | Costs | Incremental costs | DALYs | DALYs averted | ICER |
|---|---|---|---|---|---|---|---|
| *Costs and DALYs discounted at 5%* | | | | | | | |
| Baseline* | 2,106,083 | - | 32,576,457 | - | 42,289 | - | - |
| 1 (tafenoquine for adults)† | 1,731,540 | 374,543 | 42,036,795 | 9,460,338 | 32,656 | 9,632 | 982 |
| 2 (tafenoquine for all)† | 1,548,669 | 557,414 | 38,707,666 | 6,131,209 | 29,269 | 13,020 | 472 |
| 3 (high primaquine adherence)‡ | 1,891,021 | 215,062 | 45,267,047 | 12,690,589 | 35,376 | 6,912 | 1,836 |
| 4 (low primaquine adherence)§ | 1,388,891 | 717,192 | 34,985,479 | 2,409,021 | 26,651 | 15,638 | 154 |
| *Undiscounted costs and DALYs* | | | | | | | |
| Baseline* | 2,106,083 | - | 40,141,225 | - | 52,111 | - | - |
| 1 (tafenoquine for adults)† | 1,731,540 | 374,543 | 51,626,032 | 11,484,807 | 39,829 | 12,281 | 935 |
| 2 (tafenoquine for all)† | 1,548,669 | 557,414 | 47,382,476 | 7,241,251 | 35,518 | 16,592 | 436 |
| 3 (high primaquine adherence)‡ | 1,891,021 | 215,062 | 55,935,828 | 15,794,603 | 43,467 | 8,644 | 1,827 |
| 4 (low primaquine adherence)§ | 1,388,891 | 717,192 | 42,529,751 | 2,388,526 | 32,084 | 20,027 | 119 |
| *Costs and DALYs discounted at 10%* | | | | | | | |
| Baseline* | 2,106,083 | - | 27,179,528 | - | 35,282 | - | - |
| 1 (tafenoquine for adults)† | 1,731,540 | 374,543 | 35,170,590 | 7,991,062 | 27,517 | 7,765 | 1,029 |
| 2 (tafenoquine for all)† | 1,548,669 | 557,414 | 32,493,300 | 5,313,772 | 24,788 | 10,494 | 506 |
| 3 (high primaquine adherence)‡ | 1,891,021 | 215,062 | 37,646,705 | 10,467,177 | 29,596 | 5,686 | 1,841 |
| 4 (low primaquine adherence)§ | 1,388,891 | 717,192 | 29,564,278 | 2,384,750 | 22,740 | 12,541 | 190 |

*Seven-day low dose primaquine (0.5 mg/kg), primaquine adherence set at comparison scenario.

†Primaquine adherence = 66.7%.

‡Primaquine adherence = 90%.

§Primaquine adherence = 30%.

DALY, disability-adjusted life year; ICER, incremental cost-effectiveness ratio.

when children testing G6PD normal were prescribed tafenoquine (Scenario 2). With an ICER of US$154, Scenario 4 (low primaquine adherence) was the most cost-effective scenario. The smallest health gains were estimated for Scenario 3 (high primaquine adherence); however, the ICER of US$1,836 for this scenario was still below the WTP threshold. The results without discounting and with a discount rate of 10% are also provided in Table 2.

Fig 1 shows the map of ICERs by municipality. The ICER for São Gabriel da Cachoeira was US$2,145 for Scenario 1 and US$679 for Scenario 2, and the corresponding ICERs were US$1,322 and US$556 in peri-urban Manaus. In Itaituba, where occupational exposure in working-aged males drives transmission, the ICER increased to US$558 in Scenario 2 where children >2 years were prescribed tafenoquine, as compared to the ICER of US$458 for Scenario 1.

## One-way sensitivity analysis

Fig 2 shows the 10 parameter values that have the largest impact on the ICER for Scenarios 1 (tafenoquine for adults) and 2 (tafenoquine for all) in the 3 municipalities. The parameters with the largest impact on results across all municipalities were the severity and mortality due to vivax malaria, life expectancy, the lifetime and number of semiquantitative G6PD machines needed, and the costs of G6PD test strips and malaria episodes. These results also reflected transmission intensity of the municipalities. For example, Itaituba had the lowest transmission intensity so the lifetime of the semiquantitative machine, which impacts the cost per person screened, resulted in the highest impact on the ICER. The magnitude of the impact for Itaituba

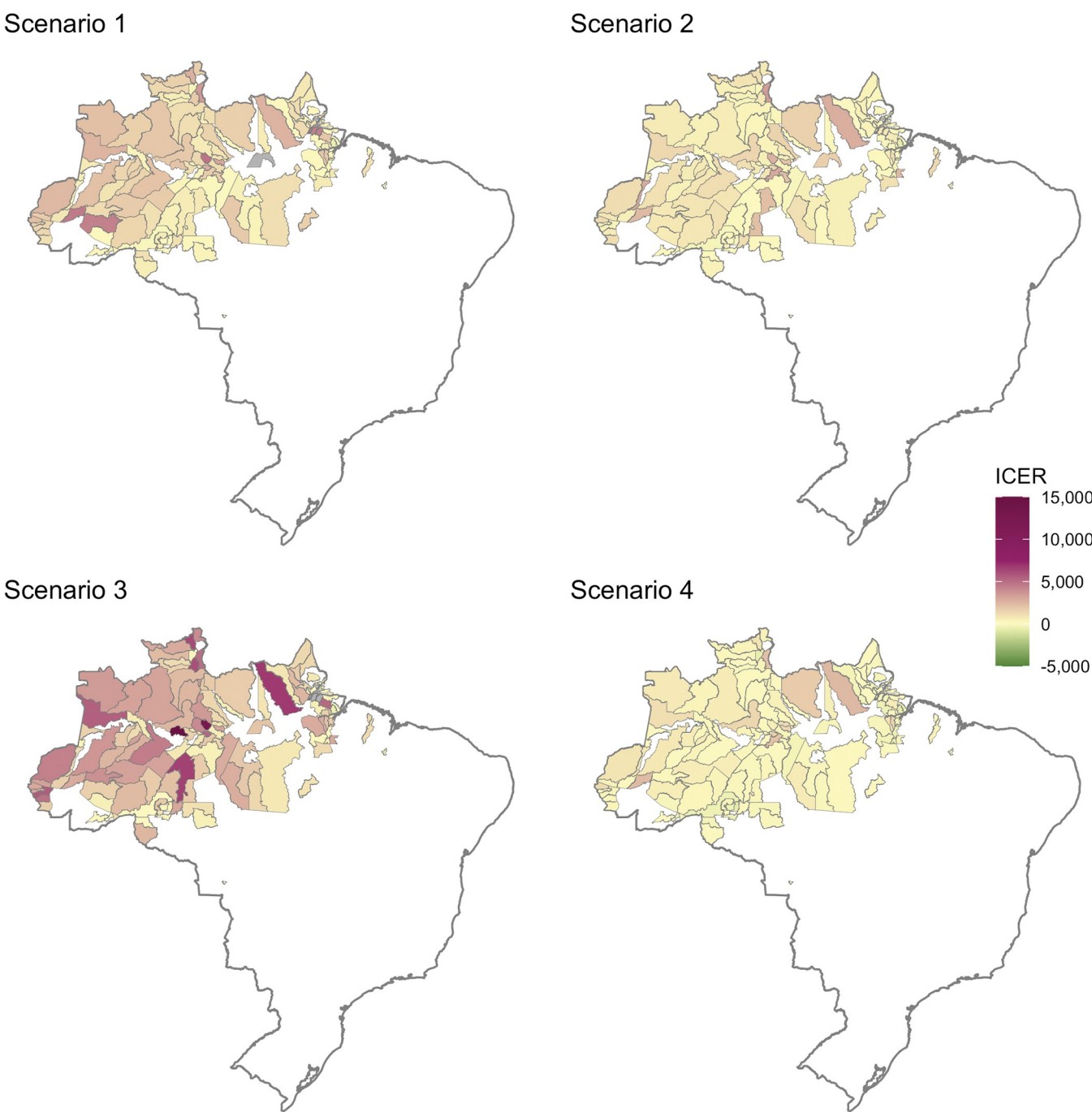

**Fig 1. Map of incremental cost-effectiveness ratios (ICERs) in each malaria-endemic municipality for Scenario 1 (tafenoquine for adults, primaquine adherence 66.7%), Scenario 2 (tafenoquine for all, primaquine adherence 66.7%), Scenario 3 (tafenoquine for adults, high primaquine adherence of 90%), and Scenario 4 (tafenoquine for adults, low primaquine adherence of 30%), compared to baseline (7 day low-dose primaquine (0.5 mg/kg), adherence set at comparison scenario).** The municipality of Manaus is plotted using results for peri-urban Manaus. The ICER for the municipality of Caapiranga for Scenario 3 was US$60,273, well above the scale. Municipalities where a strategy increased DALYs as compared to baseline are shown in grey. Maps generated with shapefiles from the R package malariaAtlas (available from: https://cran.r-project.org/web/packages/malariaAtlas/index.html).

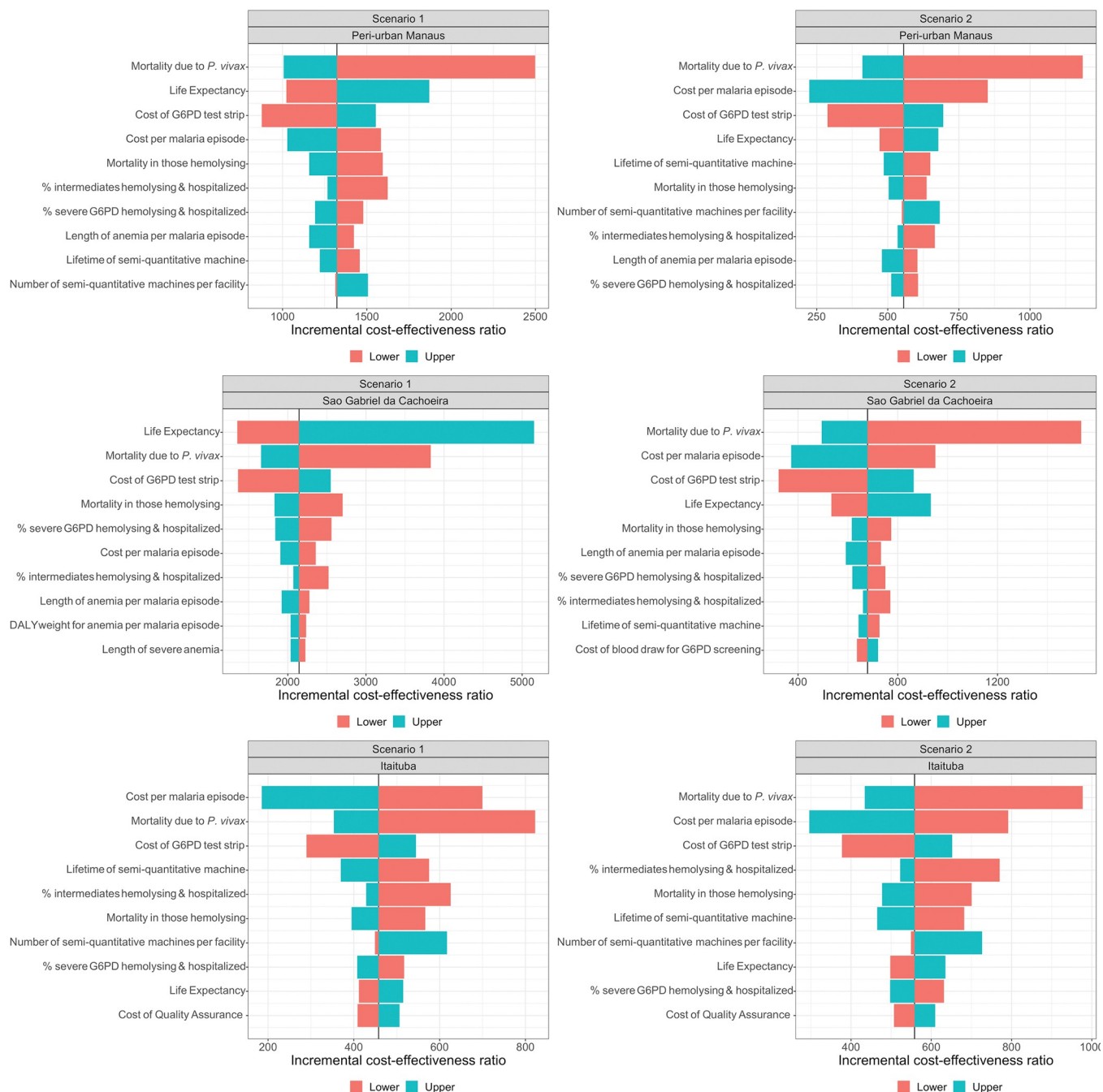

**Fig 2. One-way sensitivity analysis of the impact of changing the base case parameter value to low and high values on the incremental cost-effectiveness ratios for Scenarios 1 (tafenoquine for adults) and 2 (tafenoquine for all) as compared to the baseline (7 day low-dose primaquine (0.5 mg/kg)) for peri-urban Manaus, São Gabriel da Cachoeira, and Itaituba.** Primaquine adherence is 66.7% for all. See Table 1 for the ranges used for this analysis.

was not as large as São Gabriel da Cachoeira, which had the highest transmission intensity and for peri-urban Manaus, a moderate transmission setting. Reflecting the potential impact of tafenoquine depending on current primaquine adherence levels, the one-way sensitivity analyses in Scenario 3 (high primaquine adherence) showed a much larger impact on the results, while the impact was markedly smaller in Scenario 4 (low primaquine adherence; S2 Appendix). Scenario 3 (high primaquine adherence) was not cost-effective across all 3 municipalities.

Since occupational exposure among working-aged males drives transmission in Itaituba, the parameters with the largest impact on the results were consistent across Scenarios 1 and 2. In the other 2 municipalities where children bore more of the vivax malaria burden, the parameters with the largest impact switched around more when children were prescribed tafenoquine in Scenario 2. For Scenario 3 (high primaquine adherence), the 10% increase in life expectancy from the base case to the high value resulted in an additional 4 years for males and 5 years for females in peri-urban Manaus, while it was 6 years for males and 5 years for females in São Gabriel da Cachoeira. This had a substantial impact on the cost-effectiveness results, causing the ICER to rise beyond the WTP threshold in both municipalities (US$12,373 and US$22,467, respectively). This was driven by mortality having a larger impact on overall DALYs averted than morbidity and by high preexisting primaquine adherence decreasing the benefits in terms of relapses (and, therefore, deaths due to vivax malaria) due to prescribing tafenoquine.

## Probabilistic sensitivity analysis

S3 Appendix shows the cost-effectiveness planes produced by the PSA results. All model iterations at the country level were in the east quadrants, indicating that the tafenoquine scenarios would result in fewer DALYs than current practice. The mean ICER for Scenario 1 (tafenoquine for adults) was US$1,011 (95% CrI US$480 to US$1,837) compared to US$982 in the base case analysis, US$483 (95% CrI US$69 to US$1,052) compared to US$471 for Scenario 2 (tafenoquine for all), and US$1,954 (95% CrI US$1,104 to US$3,272) for Scenario 3 (high primaquine adherence) compared to US$1,836 (S4 Appendix). Conversely, Scenario 4 (low primaquine adherence) had decreased incremental costs while DALYs averted increased as compared to the base case analysis (S4 Appendix). While all scenarios showed higher incremental costs and DALYs averted than the base case, only Scenario 4 had a lower mean ICER with US$146 (95% CrI -US$255 to US$589) as compared to the base case (US$154). The cost-effectiveness acceptability curves in Fig 3 summarise these model iterations, by showing the percentage that fall below WTP thresholds ranging from US$0 to US$10,000. For all scenarios, 100% of model iterations were cost-effective at a WTP threshold of US$7,800. The results for the selected municipalities were similar to those at the country level, with nearly all model iterations (>99%) averting DALYs and some indicating cost savings (S5 Appendix). Again, the cost-effectiveness acceptability curves for the municipalities showed a high likelihood of being cost-effective (S6 Appendix). These results were consistent across the different transmission intensities of the selected municipalities.

## Discussion

Our results provide robust evidence that the use of tafenoquine after semiquantitative G6PD testing would be cost-effective at a WTP threshold of US$7,800, particularly in scenarios where children could be treated with a paediatric formulation or where adherence to primaquine is low. To our knowledge, this is the first cost-effectiveness analysis of tafenoquine utilising a transmission model for *P. vivax*. This analysis benefits from the robust database in Brazil (SIVEP) [23] and recent cost data collection that has occurred alongside ongoing studies in Brazil.

Despite having a large impact on its effectiveness, adherence to primaquine is challenging to estimate [30]. Unsurprisingly, adherence to primaquine had a large impact on the cost-effectiveness with the largest ICER seen for Scenario 3 (90% preexisting primaquine adherence), since single-dose tafenoquine would provide only an additional 10% improvement to adherence for radical cure in this scenario. It is reassuring that this ICER of US$1,836 in

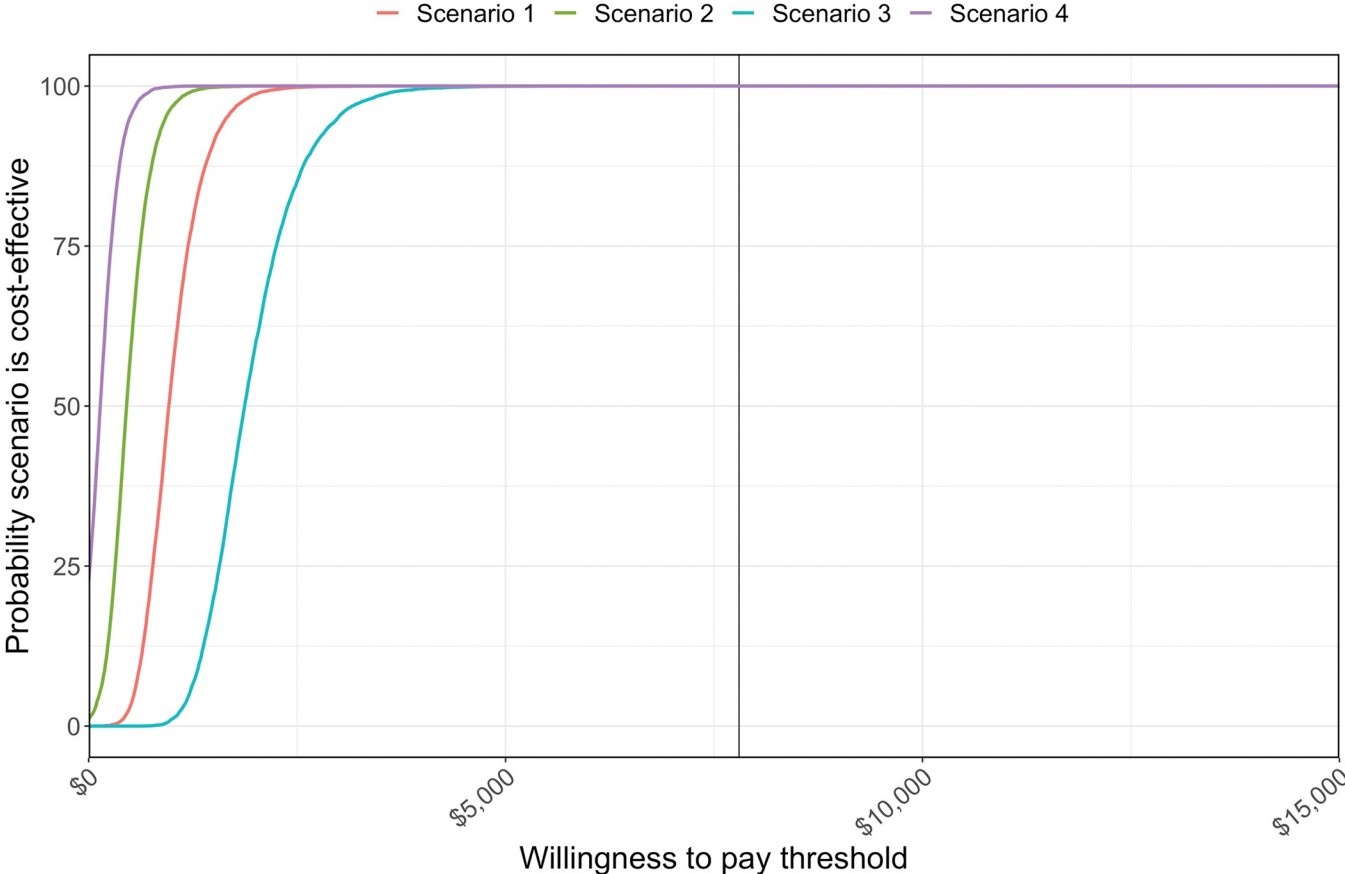

**Fig 3. Cost-effectiveness acceptability curves overall in Brazil for Scenario 1 (tafenoquine for adults, primaquine adherence 66.7%), Scenario 2 (tafenoquine for all, primaquine adherence 66.7%), Scenario 3 (tafenoquine for adults, high primaquine adherence of 90%), and Scenario 4 (tafenoquine for adults, low primaquine adherence of 30%), compared to baseline (7 day low-dose primaquine (0.5 mg/kg), adherence set at comparison scenario).** The black vertical line represents the willingness-to-pay threshold (USD$7,800).

Scenario 3 was still well below the US$7,800 WTP threshold for Brazil. A recent multicomponent intervention in Brazil resulted in similar levels of adherence; however, this level of adherence required increased investment in education, text message reminders to take primaquine, and follow-up phone surveys [31]. When assuming 30% preexisting primaquine adherence in Scenario 4, the ICER for the tafenoquine strategy had the smallest ICER (US$154), indicating improvements in health outcomes for a very small investment. While Brazil guidelines currently recommend a 7-day low-dose primaquine regimen (total dose 3.5 mg/kg), many countries continue to use the 14-day low-dose regimen (total dose 3.5 mg/kg) recommended by WHO [12], which may result in lower adherence rates than used in the base case analysis. While primaquine efficacy is high when adherence is good, our results demonstrate the potential for tafenoquine to save costs in settings where adherence to primaquine is low (30%).

The one-way sensitivity analysis on 3 municipalities with diverse epidemiology revealed that the results were least impacted by changes to parameter values in Itaituba, which had the lowest transmission intensity of the 3 municipalities examined. This is in line with the transmission model impact results, which indicated that the highest proportional drop in transmission would be in the moderate-to-low transmission settings [19]. The parameters with the largest impact on the cost-effectiveness results across all municipalities were the lifetime and number of semiquantitative G6PD machines needed, severity and mortality due to vivax

malaria, cost of G6PD test strips, and life expectancy. The lifetime of the semiquantitative G6PD machine before it needs to be replaced and number of machines needed per facility impact the total cost per person screened, so this was expected, particularly for settings with low transmission intensity like Itaituba.

A limitation of our study is regarding the uncertainty of our cost estimates for low transmission settings due to higher stochasticity of the transmission model and model assumptions. For very low transmission settings, stochastic noise and fadeout result in more unstable transmission dynamics and greater variation between simulations. In addition, calibrated incidence per 1,000 population for these low transmission settings assumes homogenous mixing in the population and no importations to sustain transmission; therefore, local transmission dynamics in communities are likely to differ compared to the simplified model aggregated municipality-level assumptions. Consequently, projected DALYs for these settings are less reliable and should be interpreted with caution. The absence of this variation in model output has minimal impact on municipalities with higher transmission (and thus at a national level) where the results converge to those from a deterministic model [19], and where the majority of the costs and benefits would be accrued. By estimating costs for a large range of transmission settings across Brazil and providing detailed results for 3 archetype settings with stable transmission, the drivers of impact on costs we identified overall are less impacted by model uncertainty.

Appropriately incorporating the costs of semiquantitative G6PD screening is challenging for a number of reasons. First, the costs of the machine, test strips, and controls have not been confirmed for procurement by the Ministry of Health in Brazil. While we have indicative costs, these may change due to distribution costs, customs, taxes, and any price changes that may occur when negotiating purchase of enough machines to implement nationally in Brazil. Second, the model assumed that G6PD screening can begin everywhere at the same time. It is likely that the rollout will occur gradually and that uptake may be slow or patchy. In addition, it is assumed that a semiquantitative machine will be placed at health units that had a case of vivax malaria in 2018 throughout the entire time horizon of the analysis. For large units and units that make home visits, more than one machine might be needed; this would increase the costs. Alternatively, health units that do not continue to see malaria cases may not need to continue stocking a semiquantitative machine. Finally, since transmission is an important driver of results, these findings are dependent on future trends in vivax malaria cases.

Severity and mortality of vivax malaria are also challenging parameters to estimate as data are sparse. The base case value of 0.03% used here was from a study of vivax malaria patients admitted to a hospital in Brazil during 2009 to 2011 [13] and from a review that accessed 2014 data on microscopically confirmed malaria cases and related deaths from the National Malaria Prevention and Control Programme, Ministry of Health of Brazil [32]. The latter included falciparum malaria, indicating that it may be an overestimate. Another study of vivax malaria from a tertiary care centre in Manaus from 1996 to 2010 found a lower case fatality rate of 0.01% [15]. To be conservative, our low value for the sensitivity analyses assumed no mortality due to vivax malaria. While this parameter had a large impact on the results in the one-way sensitivity analysis, all scenarios remained cost-effective across all 3 municipalities when this assumption was applied.

Finally, this only compares tafenoquine with low-dose primaquine treatment. While low-dose primaquine (3.5 mg/kg total) is the current recommended treatment in Brazil, a recent comparison of low-dose with a high-dose primaquine regimen (7.0 mg/kg total) found a 27% difference in the percentage of patients who were recurrence-free at day 168 when these regimens were supervised [33]. While tafenoquine has been shown to have similar efficacy to low-dose primaquine [10], it has not been directly compared to high-dose primaquine in Brazil.

This clinical comparison would need to be done before the implications for the cost-effectiveness analysis could be ascertained.

## Conclusions

Our cost-effectiveness analysis using a transmission model calibrated to epidemiological data from Brazil demonstrates a high probability of tafenoquine to be cost-effective at a threshold of US$7,800 per DALY averted, following a normal test result with a semiquantitative G6PD test. This intervention is most likely to be cost-effective in situations where primaquine adherence is low and when paediatric formulations enable it to be prescribed to children over the age of 6 months.

## Supporting information

**S1 CHEERS Checklist. Consolidated Health Economic Evaluation Reporting Standards (CHEERS) guidelines.**
(DOCX)

**S1 Appendix. Economic evaluation model code.**
(R)

**S2 Appendix. One-way sensitivity analysis of the impact of changing the base case parameter value to low and high values on the incremental cost-effectiveness ratios (ICERs) for Scenarios 3 (high primaquine adherence) and 4 (low primaquine adherence) as compared to the baseline scenario for peri-urban Manaus, São Gabriel da Cachoeira, and Itaituba.** See Table 1 for the ranges used for this analysis.
(TIF)

**S3 Appendix. Cost-effectiveness planes showing the incremental costs and disability-adjusted life years (DALYs) averted from 10,000 iterations in the probabilistic sensitivity analysis.** Results are for Scenario 1 (tafenoquine for adults, primaquine adherence 66.7%), Scenario 2 (tafenoquine for all, primaquine adherence 66.7%), Scenario 3 (tafenoquine for adults, high primaquine adherence of 90%), and Scenario 4 (tafenoquine for adults, low primaquine adherence of 30%) compared to baseline (7-day low-dose primaquine (0.5 mg/kg), adherence set at comparison scenario) overall for Brazil. The base case analysis results are designated by a black triangle in each panel.
(TIF)

**S4 Appendix. Mean and 95% credible intervals from the probabilistic sensitivity analysis for disability-adjusted life-years (DALYs) averted, incremental costs, and incremental cost effectiveness ratios (ICERs).**
(DOCX)

**S5 Appendix. Cost-effectiveness planes showing the incremental costs and disability-adjusted life years (DALYs) averted from 10,000 iterations in the probabilistic sensitivity analysis.** All scenarios are compared to the baseline scenario for 3 municipalities. The base case analysis results are designated by a black triangle in each panel.
(TIF)

**S6 Appendix. Cost-effectiveness acceptability curves for each tafenoquine scenario in 3 municipalities compared to the baseline scenario.** The black vertical line represents the willingness-to-pay threshold (USD$7,800).
(TIF)

## Author Contributions

**Conceptualization:** Narimane Nekkab, Wuelton M. Monteiro, Michael T. White, Angela Devine.

**Data curation:** Daniel A. M. Villela.

**Formal analysis:** David J. Price, Narimane Nekkab, Angela Devine.

**Funding acquisition:** Angela Devine.

**Methodology:** David J. Price, Narimane Nekkab, Wuelton M. Monteiro, Daniel A. M. Villela, Marcus V. G. Lacerda, Angela Devine.

**Supervision:** Julie A. Simpson, Marcus V. G. Lacerda, Michael T. White, Angela Devine.

**Writing – original draft:** David J. Price, Angela Devine.

**Writing – review & editing:** Narimane Nekkab, Wuelton M. Monteiro, Julie A. Simpson, Marcus V. G. Lacerda, Michael T. White.

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
