## [Editor Report · Decision Letter 0]

1 Jun 2023

Dear Dr Devine, 

Thank you for submitting your manuscript entitled "Tafenoquine following G6PD screening versus primaquine for the treatment of vivax malaria in Brazil: a cost-effectiveness analysis using a transmission model" for consideration by PLOS Medicine.

Your manuscript has now been evaluated by the PLOS Medicine editorial staff as well as by an academic editor with relevant expertise and I am writing to let you know that we would like to send your submission out for external peer review.

Please re-submit your manuscript within two working days, i.e. by Jun 05 2023 11:59PM.

Kind regards,

Katrien Janin, PhD

Senior Editor

PLOS Medicine

---

## [Decision Letter · Decision Letter 1]

1 Oct 2023

Dear Dr. Devine,

Thank you very much for submitting your manuscript "Tafenoquine following G6PD screening versus primaquine for the treatment of vivax malaria in Brazil: a cost-effectiveness analysis using a transmission model" (PMEDICINE-D-23-01515R1) for consideration at PLOS Medicine. 

We do apologise for the delay in you receiving feedback on your manuscript. Your paper was evaluated by a senior editor and discussed among all the editors here. It was also discussed with an academic editor with relevant expertise, and sent to independent reviewers, including a statistical reviewer. The reviews are appended at the bottom of this email and any accompanying reviewer attachments can be seen via the link below:

[LINK]

In light of these reviews, we are not be able to accept the manuscript for publication in the journal in its current form, but we would like to consider a revised version that addresses the reviewers' and editors' comments. We cannot make any decision about publication until we have seen the revised manuscript and your response, and we plan to seek re-review by one or more of the reviewers. 

We expect to receive your revised manuscript by Oct 20 2023 11:59PM. Please email us (plosmedicine@plos.org) if you have any questions or concerns.

We look forward to receiving your revised manuscript. 

Sincerely,

Katrien Janin, PhD

PLOS Medicine

plosmedicine.org

Comment from the Academic Editor:

The reviewers were generally positive about the paper and its suitability for PLOS Medicine. They provided a number of suggestions for the authors to address in their revision and I believe the clarifications, corrections, and further information would improve the paper or be valuable information for future readers of the paper.

GENERAL:

Your manuscript has been assessed by four reviewers whose reports can be found below. As you will see from the comments, the reviewers have raised a number of concerns that need addressing. Please carefully revise the manuscript to address all comments raised.

For in-text reference, citations are placed within square parentheses and should precede punctuation as follows, e.g see line 29 “…. 17.6 million United States Dollars (US$) in 2017 [3].’ Please check and amend throughout. 

Please provide 95% CIs and p values for all results were appropriate, check and amend throughout. For p values, please report these as p<0.001 and where higher as p=0.002 or p=0.050. Suggest reporting statistical information for clarity in the following format: ‘x’; (95% CI [‘y’,’ z’] p<0.001).

STUDY DESIGN:

i) For the economic evaluation part of your manuscript, please report your economic analysis according to the appropriate study design provided at http://www.equator-network.org/?post_type=eq_guidelines&eq_guidelines_study_design=economic-evaluations&eq_guidelines_clinical_specialty=0&eq_guidelines_report_section=0&s= and provide the relevant completed checklist. In the checklist please include sufficient text excerpted from the manuscript to explain how you accomplished all applicable items.

ii) For the transmission model, we ask for inclusion of specific items, derived from Geoffrey P Garnett, Simon Cousens, Timothy B Hallett, Richard Steketee, Neff Walker. Mathematical models in the evaluation of health programmes. (2011) Lancet DOI:10.1016/S0140-6736(10)61505-X.

Please ensure all the items listed below are included with your manuscript. Please review the list below and confirm/revise as necessary: 

(i) Please provide a diagram that shows the model structure, including how the disease natural history is represented, the process and determinants of disease acquisition, and how the putative intervention could affect the system.

(ii) Please provide a complete list of model parameters, including clear and precise descriptions of each parameter, together with the values or ranges for each, with justification or the primary source cited, and important caveats about the use of these values noted.

(iii) Please provide a clear statement about how the model was fitted to the data.

(iv) For uncertainty analyses, please state the sources of uncertainties quantified and not quantified.

(v) Please provide sensitivity analyses to identify which parameter values are most important in the model. Uncertainty estimates seek to derive a range of credible results on the basis of an exploration of the range of reasonable parameter values. The choice of method should be presented and justified.

(vi) Please discuss the scientific rationale for this choice of model structure and identify points where this choice could influence conclusions drawn. Please also describe the strength of the scientific basis underlying the key model assumptions.

ABSTRACT:

For the abstract, please structure the abstracts along the following 3 headings: Background, Methods and Findings, Conclusions. (Currently there are separate headings for Methods and Findings sections). Please remove all other subheaders.

Abstract Background: Provide the context of why the study is important. The final sentence should clearly state the study question.

Abstract Methods and Findings:

Please quantify the main results with 95% CIs and p values. In the last sentence of the Abstract Methods and Findings section, please describe the main limitation(s) of the study's methodology.

Abstract Conclusions:

Please interpret the study based on the results presented in the abstract, emphasizing what is new without overstating your conclusions.

AUTHORS SUMMARY:

Ideally each sub-heading should contain 2-3 single sentence, concise bullet points containing the most salient points from your study.

In the final bullet point of ‘What Do These Findings Mean?’ Please include the main limitations of the study in non-technical language.

ACKNOWLEDGMENTS/ DECLARATIONS

Please remove all statements apart from acknowledgements, author contributions and abbreviations from the end of the main manuscript and include these only in the relevant parts of the manuscript submission form. Funding, competing interest, and data availability will be compiled as metadata. The ethics declaration should be included in your method section. 

Comments from the reviewers:

Reviewer #1: This is a well-written cost-effectiveness study that makes a single point in an important country for relapsing malaria. Tafenoquine is a better choice (more cost effective) than reasonable assumptions for the current primaquine regimen. As the authors mention in the discussion, the analysis shifts when higher dose primaquine regimens are used, but so do the risks. This study should encourage the Brazilian government to continue its phased roll out of tafenoquine to eliminate vivax malaria in hopes that further practical information will be gained from actual field data for use in other countries / situations. PLOS states that it also gets statistical reviews for its papers, which is good as although the numbers seem reasonable to me, they need to be examined by an outside statistician. 

Reviewer #2: This is an innovative and interesting article which uses mixed methods to provide robust input to a proven model. It provides guidance as to the use of new technology that seems appropriate for policy action. I think it would help the reading public to have a bit more explanation of two areas that are mentioned but to my mind could be more complete. First a more complete review of the semi-quantitative device utilized including country of origin and cost

as well as explicit use parameters i.e. training needed and if there are other requirements to make it operational. Introducing new technology is always a tricky process and the application of implementation science principles might be suggested. 

A second observation that could greatly impact implementation is the mentioned compliance with treatment once cases have been identified. This may require more costly behavioral change programs which do not seem to be reflected in the model. Perhaps a suggestion as to how roll out might be effected given model results and use would be useful to policy makers. 

In all a good and interesting article worthy of publication.

Reviewer #3: The authors have produced a clear and well written paper to assess the cost effectiveness of rolling out Tafenoquine following G6PD screening vs primaquine for the treatment of vivax malaria in Brazil. 

There are three minor comments:

1. While the dynamic transmission model paper is published elsewhere, because the CEA is based on it, it is necessary for the reader to understand the assumptions made for future interventions in the baseline scenario. The transmission model baseline projections should be described in the methods section. This will provide the reader with context for understanding geographical differences in CEA results. 

2. No information was given on treatment access and how it may differ geographically. If the scenarios assume passive treatment only, then differences in treatment access will impact the analysis. 

3. A section should be added to the discussion on the key sources of variation in the dynamic transmission model and how it impacts the CEA sensitivity analysis and results. It was unclear whether the n random draws from the cost parameter distributions were applied to m transmission model baseline simulations to result in n x m iterations. Please elucidate. 

Reviewer #4: See attachment

Michael Dewey

[LINK]

---

## [Decision Letter · Decision Letter 2]

28 Nov 2023

Dear Dr. Devine,

Thank you very much for re-submitting your manuscript "Tafenoquine following G6PD screening versus primaquine for the treatment of vivax malaria in Brazil: a cost-effectiveness analysis using a transmission model" (PMEDICINE-D-23-01515R2) for review by PLOS Medicine.

I have discussed the paper with my colleagues and the academic editor and it was also seen again by the reviewers. I am pleased to say that provided the remaining editorial and production issues are dealt with we are planning to accept the paper for publication in the journal.

[LINK]

We expect to receive your revised manuscript within 1 week. Please email us (plosmedicine@plos.org) if you have any questions or concerns, or like an extension. 

We ask every co-author listed on the manuscript to fill in a contributing author statement. If any of the co-authors have not filled in the statement, we will remind them to do so when the paper is revised. If all statements are not completed in a timely fashion this could hold up the re-review process. Should there be a problem getting one of your co-authors to fill in a statement we will be in contact. 

If you have any questions in the meantime, please contact me or the journal staff on plosmedicine@plos.org. For editorial questions, please feel free to contact me directly on kjanin@plos.org 

We look forward to receiving the revised manuscript by Dec 05 2023 11:59PM.   

Sincerely,

Katrien Janin, PhD

Senior Editor 

PLOS Medicine

plosmedicine.org

Comments from the Editors:

Thank you for fully addressing our previous comments

We just have a very minor remark, regarding references.

Please use the "Vancouver" style for reference formatting, and see our website for other reference guidelines https://journals.plos.org/plosmedicine/s/submission-guidelines#loc-references

For online references (e.g see reference 26 and 27), please use the following format: [accessed on 2 Apr 2021] instead of [cited 2020 10 Nov]. For reference 26, I noticed the presence of a double ‘;’ 

Please double check your references and amend as needed.

We having given 5 working days to return the manuscript to us, but in your case please feel free to submit much sooner if that suits you. We look forward to issuing the editorial acceptance for you.

Comments from Reviewers:

Reviewer #2: Much improved version which seems to respond to all of the reviewer comments.. i did notice that there was no reference to IRB approval which given the nature of the research may be acceptable i.e. secondary data etc.

i believe that the researchers could constructively add a paragraph on how these results might be interpreted by ph practitioners and or what other complementary research would be needed and or beneficial to using the results.

Reviewer #4: The authors have addressed all my points.

Michael Dewey

[LINK]

---

## [Editor Report · Decision Letter 3]

29 Nov 2023

Dear Dr Devine, 

On behalf of my colleagues, I am pleased to inform you that we have agreed to publish your manuscript "Tafenoquine following G6PD screening versus primaquine for the treatment of vivax malaria in Brazil: a cost-effectiveness analysis using a transmission model" (PMEDICINE-D-23-01515R3) in PLOS Medicine.

Thank you again for submitting to PLOS Medicine. We look forward to publishing your paper !

Sincerely, 

Katrien G. Janin, PhD 

Senior Editor 

PLOS Medicine